# Towards hyperparameter-free optimization with differential privacy

**Ruixuan Liu** [*]
Emory University
ruixuan.liu2@emory.edu

**Zhiqi Bu** [*]
Amazon
woodyx218@gmail.com

## Abstract

Differential privacy (DP) is a privacy-preserving paradigm that protects the training data when training deep learning models. Critically, the performance of models is determined by the training hyperparameters, especially those of the learning rate schedule, thus requiring fine-grained hyperparameter tuning on the data. In practice, it is common to tune the learning rate hyperparameters through the grid search that (1) is computationally expensive as multiple runs are needed, and (2) increases the risk of data leakage as the selection of hyperparameters is data-dependent. In this work, we adapt the automatic learning rate schedule to DP optimization for any models and optimizers, so as to significantly mitigate or even eliminate the cost of hyperparameter tuning when applied together with automatic per-sample gradient clipping. Our hyperparameter-free DP optimization is almost as computationally efficient as the standard non-DP optimization, and achieves state-of-the-art DP performance on various language and vision tasks.

## 1 Introduction

The performance of deep learning models relies on a proper configuration of training hyperparameters. In particular, the learning rate schedule is critical to the optimization, as a large learning rate may lead to divergence, while a small learning rate may slowdown the converge too much to be useful. In practice, people have used heuristic learning rate schedules that are controlled by many hyperparameters. For example, many large language models including LLaMa2 (Touvron et al., 2023) uses linear warmup and cosine decay in its learning rate schedule, which are controlled by 3 hyperparameters. Generally speaking, hyperparameter tuning (especially for multiple hyperparameters) can be expensive for large datasets and large models.

To address this challenge, it is desirable or even necessary to determine the learning rate schedule in an adaptive and data-dependent way, without little if any manual effort. Recent advances such as D-adaptation (Defazio & Mishchenko, 2023), Prodigy (Mishchenko & Defazio, 2024), DoG (Ivgi et al., 2023), DoWG (Khaled et al., 2023), U-DoG (Kreisler et al., 2024), and GeN (Bu & Xu, 2024) have demonstrated the feasibility of automatic learning rate schedule, with some promising empirical results in deep learning (see a detailed discussion in Section 2.3).

While data-dependent learning rate are evolving in the standard non-DP regime, the adaptive data dependency can raise the privacy risks of memorizing and reproducing the training data. As an example, we consider the zeroth-order optimizer (ZO-SGD),

$$\boldsymbol{w}_{t+1} = \boldsymbol{w}_t - \eta_{\text{ZO-SGD}} \boldsymbol{z}_t$$

where $\boldsymbol{w}_t$ is the model parameters, $\boldsymbol{z}_t$ is a random vector that is independent of any data, and $\eta_{\text{ZO-SGD}} \approx \frac{\partial L}{\partial \boldsymbol{w}_t}^\top \boldsymbol{z}_t \in \mathbb{R}$ is the effective learning rate, with $L$ being the loss value. In Malladi et al. (2023), ZO-SGD can effectively optimize models such as OPT 13B$\sim$60B in the few-shot and many-shot settings. Hence, by using an adaptive hyperparameter $\eta_{\text{ZO-SGD}}$, the models are able to learn and possibly memorize the training data even if the descent direction is data-independent. In fact, the private information can also leak through other hyperparameters and in other models,

---

[*]Equal contribution. This work does not relate to ZB's position at Amazon.

where an outlier datapoint can be revealed via the membership inference attacks in support vector machines (Papernot & Steinke, 2021).

Specifically, in the regime of deep learning with differential privacy (DP), the privacy risks from the non-DP hyperparameter tuning could render the privacy guarantee non-rigorous. In practice, most work has leveraged the DP optimizers Abadi et al. (2016) to offer the privacy guarantee, where the privatization is only on the gradients, but not on the hyperparameters such as the clipping threshold $R_g$ and the learning rate $\eta$. A common approach is to trial-and-error on multiple $(R_g, \eta)$ pairs and select the best hyperparameters, as showcased by (Li et al., 2021; Kurakin et al., 2022) in Figure 8.

At high level, there are two approaches to accommodate the privacy risk in hyperparameter tuning: (1) The more explored approach is to assign a small amount of privacy budget to privatize the hyperparameter tuning. Examples include Renyi-DP tuning (Papernot & Steinke, 2021), DP-Hypo (Wang et al., 2023), DP-ZO-SGD (Tang et al., 2024; Liu et al., 2024; Zhang et al.) and many more (Liu & Talwar, 2019; Panda et al., 2024; Koskela & Kulkarni, 2023). However, these methods may suffer from worse performance due to larger DP noise addition from the reduced privacy budget, or high computation overhead. (2) The less explored approach is to adopt hyperparameter-free methods. For instance, (Bu et al., 2023b; Yang et al., 2022) replace the per-sample gradient clipping with the per-sample gradient normalization (i.e. setting $R_g \approx 0^+$), so as to remove the tuning of the hyperparameter $R_g$ in DP optimizers.

In this work, we work towards the hyperparameter-free optimization with DP, by adapting learning-rate-free methods in DP optimization:

$$\text{DP-SGD} \Longrightarrow \begin{cases} \text{Vanilla:} & \boldsymbol{w}_{t+1} = \boldsymbol{w}_t - \eta_{\text{manual}} \left( \sum_{i \in \mathcal{B}} \min\{\frac{R_g}{||\boldsymbol{g}_i||}, 1\} \boldsymbol{g}_i + \sigma R_g \boldsymbol{z} \right) \\ \text{Hyperparameter-free:} & \boldsymbol{w}_{t+1} = \boldsymbol{w}_t - \eta_{\text{GeN-DP}} \left( \sum_{i \in \mathcal{B}} \frac{\boldsymbol{g}_i}{||\boldsymbol{g}_i||} + \sigma \boldsymbol{z} \right) \end{cases}$$

(1)

Our contributions are summarized as follows:

1. We propose **HyFreeDP**, a hyperparameter-free DP framework that rigorously guarantees DP on the hyperparameter tuning of $(R_g, \eta_t)$, and works with any optimizer.

2. We apply the loss privatization to leverage GeN learning rate under DP, with a specific auto-regressive clipping threshold $R_l$ that aims to minimize the clipping bias.

3. We give an end-to-end privacy accounting method that adds $< 1\%$ more gradient noise, while accurately capturing the loss curvature to determine the adaptive learning rate.

4. We show the strong performance and high efficiency of our method empirically.

## 2 PRELIMINARIES AND RELATED WORKS

### 2.1 DIFFERENTIALLY PRIVATE OPTIMIZATION

**Overview of DP optimization.** We aim to minimize the loss $\sum_i L(\boldsymbol{w}, \boldsymbol{x}_i)$ where $\boldsymbol{w} \in \mathbb{R}^d$ is the model parameters and $\boldsymbol{x}_i$ is one of data points with $1 \leq i \leq N$. We denote the per-sample gradient as $\boldsymbol{g}_i(\boldsymbol{w}) := \frac{\partial L(\boldsymbol{w}, \boldsymbol{x}_i)}{\partial \boldsymbol{w}} \in \mathbb{R}^d$ and the mini-batch gradient at the $t^{th}$ updating iteration as $\boldsymbol{m}_t \in \mathbb{R}^d$: for a batch size $B \leq N$,

$$\boldsymbol{m}_t(\{\boldsymbol{g}_i\}_{i=1}^B; R_g, \sigma_g) := [\sum_i^B c_i(R_g) \boldsymbol{g}_i + \sigma_g R_g \cdot \boldsymbol{z}_g]/B \tag{2}$$

where $\boldsymbol{z}_g \sim \mathcal{N}(0, \mathbf{I}_d)$ is the Gaussian noise, and $c_i(R_g) = \min(R_g/||\boldsymbol{g}_i||, 1)$ is the per-sample gradient clipping factor in (Abadi et al., 2016; De et al., 2022) with $R_g$ being the clipping threshold.

In particular, $\boldsymbol{m}_t$ reduces to the standard non-DP gradient $\sum_i \boldsymbol{g}_i/B$ when $\sigma_g = 0$ and $R_g$ is sufficiently large (i.e., no clipping is applied). The clipped and perturbed batch gradient becomes the DP gradient $\boldsymbol{m} \equiv \boldsymbol{m}_{\text{DP}}$ whenever $\sigma_g > 0$, with stronger privacy guarantee for larger $\sigma_g$. On top of $\boldsymbol{m}$, models can be optimized using any optimizer such as SGD and AdamW through

$$\boldsymbol{w}_{t+1} = \boldsymbol{w}_t - \eta_t \cdot \mathbf{G}_t(\boldsymbol{m}_t(\{\boldsymbol{g}_i\}_{i=1}^B)) \tag{3}$$

in which $\mathbf{G}_t$ is the post-processing such as momentum, adaptive pre-conditioning, and weight decay.

**Hyper-parameters matter for DP optimization.** Previous works (De et al., 2022; Li et al., 2021) reveal that the performance of DP optimization is sensitive to the hyper-parameter choices. On the one hand, DP by itself brings extra hyper-parameters, such as the gradient clipping threshold $R_g$, making the tuning more complex. An adaptive clipping method (Andrew et al., 2021) proposes to automatically learn $R_g$ at each iteration, with an extra privacy budget that translates to worse accuracy. There are methods that do not incur extra privacy budget. Automatic clipping (or per-sample normalization) (Bu et al., 2023b; Yang et al., 2022) uses $c_i = 1/||\boldsymbol{g}_i||$ to replace the $R_g$-dependent per-sample clipping, and thus removes the hyperparameter $R_g$ from DP algorithms. On the other hand, hyper-parameter tuning for DP training has different patterns than non-DP training and cannot borrow previous experience on non-DP. For example, previous works (Li et al., 2021; De et al., 2022) observe that there is no benefit from decaying the learning rate during training and the optimal learning rate can be much higher than the optimal one in non-DP training.

## 2.2 END-TO-END DP GUARANTEE FOR OPTIMIZATION AND TUNING

However, hyper-parameter tuning enlarges privacy risks (Papernot & Steinke, 2021), and it is necessary to provide end-to-end privacy guarantee for DP. There are two existing technical paths to solve this problem:

- Multiple runs with multiple choices of hyper-parameters, and choose from these choices in a DP manner (Mohapatra et al., 2022; Papernot & Steinke, 2021; Wang et al., 2023; Liu & Talwar, 2019). This path requires some knowledge to determine the choices a priori and consumes a significant privacy budget as well as computation due to the multiple runs. For instance, (Liu & Talwar, 2019) showed that repeated searching with random iterations satisfies $(3\epsilon, 0)$-DP if each run was $(\epsilon, 0)$-DP.

- Making DP optimization hyper-parameter free, and only a single run is needed. For example, the automatic clipping (Bu et al., 2023b; Yang et al., 2022) eliminates the hyper-parameter $R_g$. Our work follows the second path, aiming to resolve tuning on $\eta$—the last critical hyper-parameter—by finding a universal configuration across datasets and tasks.

## 2.3 AUTOMATIC LEARNING RATE SCHEDULE

Automatic learning rate schedules (also known as learning-rate-free or parameter-free methods) have demonstrated promising performance in deep learning, with little to none manual efforts to select the learning rate. Specifically, D-adaptation (Defazio & Mishchenko, 2023), Prodigy (Mishchenko & Defazio, 2024), and DoG (Ivgi et al., 2023) (and its variants) have proposed to estimate $\eta \approx \frac{D}{G\sqrt{T}}$ where $D = ||\boldsymbol{w}_0 - \boldsymbol{w}_*||$ is the initialization-to-minimizer distance, $G$ is the Lipschitz continuity constant, and $T$ is the total number of iterations. These methods have their roots in the convergence theory under the convex and Lipschitz conditions, and may not be accurate when applied in the DP regime when the gradient is noisy. In fact, the estimation of $D$ and $G$ can deviate significantly from the truth when $\epsilon$ is small and the number of trainable parameters is large, i.e. the DP noise in gradient is large, as shown in Table 2.

Along an orthogonal direction, GeN (Bu & Xu, 2024) leverages the Taylor approximation of loss to set the learning rate $\eta_{\text{GeN}}$, without assuming the Lipschitz continuity or the knowledge of $D$. Given any descent vector $\boldsymbol{m}$, omitting the iteration index $t$ for a brief notation, we can use the GeN learning rate in (4) to approximately minimize $L$:

$$\eta_{\text{GeN}}(\boldsymbol{m}) := \frac{\mathbf{G}^\top \boldsymbol{m}}{\boldsymbol{m}^\top \mathbf{H} \boldsymbol{m}} = \text{argmin}_\eta L(\boldsymbol{w}) - \mathbf{G}^\top \boldsymbol{m}\eta + \boldsymbol{m}^\top \mathbf{H}\boldsymbol{m}\frac{\eta^2}{2} \approx \text{argmin}_\eta L(\boldsymbol{w} - \eta\boldsymbol{m}) \quad (4)$$

in which $\mathbf{G} = \frac{\partial L}{\partial \boldsymbol{w}}$ is the gradient and $\mathbf{H} = \frac{\partial^2 L}{\partial \boldsymbol{w}^2}$ is the Hessian matrix. Notice that because $L$ is approximated by a quadratic function, the minimizer $\eta_{\text{GeN}}$ is unique and in closed form.

Numerically, $\eta_{\text{GeN}}$ can be computed up to any precision by curve fitting or finite difference. Under the non-DP regime, given a series of $\eta_i$ and loss values $L(\boldsymbol{w} - \eta_i\boldsymbol{m})$, Bu & Xu (2024) obtains the

numerator and denominator of $\eta_{\text{GeN}}$ by solving the problem in (5):

$$\boldsymbol{m}^\top \mathbf{H} \boldsymbol{m}, \mathbf{G}^\top \boldsymbol{m} \approx \operatorname{argmin}_{a,b} \sum_i \left| L(\boldsymbol{w} - \eta_i \boldsymbol{m}) - \left( L(\boldsymbol{w}) - b\eta_i + a\frac{\eta_i^2}{2} \right) \right|^2 \tag{5}$$

Nevertheless, directly applying these non-DP automatic learning rate scheduler with DP gradient in (2) and using $\eta_{\text{GeN}}(\boldsymbol{m})$ will violate DP, because the learning rate estimation is obtained from forward passes on batches of private data. We defer the explanation and solution to Section 3.

## 3 LOSS VALUE PRIVATIZATION WITH MINIMAL CLIPPING BIAS

### 3.1 PRIVATIZED QUADRATIC FUNCTION

We emphasize that the learning rate $\eta_{\text{GeN}}(\boldsymbol{m})$ is not DP, because even though $\boldsymbol{m}$ is privatized, the data is accessed without protection through $\mathbf{G}$ and $\mathbf{H}$. To solve this issue, we introduce a privatized variant of (5),

$$(\boldsymbol{m}^\top \mathbf{H} \boldsymbol{m})_{\text{DP}}, (\mathbf{G}^\top \boldsymbol{m})_{\text{DP}} := \operatorname{argmin}_{a,b} \sum_i \left| \tilde{L}(\boldsymbol{w} - \eta_i \boldsymbol{m}_{\text{DP}}) - \left( \tilde{L}(\boldsymbol{w}) - b\eta_i + a\frac{\eta_i^2}{2} \right) \right|^2 \tag{6}$$

which not only replaces $\boldsymbol{m}$ with the DP gradient $\boldsymbol{m}_{\text{DP}}$ but also privatizes the loss by $\tilde{L}(\boldsymbol{w} - \eta_i \boldsymbol{m}_{\text{DP}})$ as in (7). The resulting learning rate is $\eta_{\text{GeN-DP}} = \frac{(\mathbf{G}^\top \boldsymbol{m})_{\text{DP}}}{(\boldsymbol{m}^\top \mathbf{H} \boldsymbol{m})_{\text{DP}}}$, which is DP since every quantity in (6) is DP and because of the post-processing property. We now discuss the specifics of the loss value privatization $\tilde{L}(\boldsymbol{w} - \eta_i \boldsymbol{m}) \in \mathbb{R}$. In Table 1, we emphasize that the loss privatization is distinctively different from the gradient privatization because the loss is scalar, whereas the gradient is high-dimensional.

Table 1: Difference between the privatization of loss and gradient.

| Aspects | Loss Privatization | Gradient Privatization |
|---|---|---|
| Dimension | 1 | $d$ |
| Clipping Norm | L2 or L1 | L2 |
| Noise Magnitude | $\sqrt{\frac{2}{\pi}} \frac{\sigma_l R_l}{B}$ | $\sqrt{d} \frac{\sigma_g R_g}{B}$ |
| Key to Convergence | Clipping Bias | Noise Magnitude |
| Per-sample Operation | Clipping ($R_l \approx L$) | Normalization ($R_g \approx 0^+$) |

From the perspective of per-sample clipping, the gradient is ubiquitously clipped on L2 norm, because $\|\boldsymbol{g}_i\|_2 \ll \|\boldsymbol{g}_i\|_1$ in large neural networks, and consequently a Gaussian noise is added to privatize the gradient. In contrast, we can apply L2 or L1 norm for the loss clipping, and add Gaussian (by default) or Laplacian noise to the loss, respectively.

From the perspective of noising, the expected noise magnitude for loss privatization is $\mathbb{E}|z_l| \frac{\sigma_l R_l}{B} = \sqrt{\frac{2}{\pi}} \frac{\sigma_l R_l}{B}$ where $z_l \sim N(0,1)$ is the noise on loss, and that for gradient privatization is $\mathbb{E}\|\boldsymbol{z}_g\| \frac{\sigma_g R_g}{B} \approx \sqrt{d} \frac{\sigma_g R_g}{B}$ where the gradient noise vector $\boldsymbol{z}_g \sim N(0, \mathbf{I}_d)$ by the law of large numbers. On the one hand, the gradient noise $\boldsymbol{z}_g$ is large and requires small $R_g$ to suppress the noise magnitude, as many works have use very small $R$ (Li et al., 2021; De et al., 2022). This leads to the automatic clipping in Bu et al. (2023b) when the gradient clipping effectively becomes the gradient normalization as $R_g \to 0^+$. In fact, each per-sample gradient (as a vector) has a magnitude and a direction, and the normalization neglects some if not all magnitude information about the per-sample gradients. On the other hand, we must not use a small $R_l$ for the loss clipping because the per-sample loss (as a scalar) only has the magnitude information. We will show by Theorem 1 that the choice of threshold $R_l$ creates a bias-variance tradeoff between the clipping and the noising for the loss privatization:

$$\tilde{L} = \frac{1}{B} \left[ \sum_i \min(\frac{R_l}{L_i}, 1) L_i + \sigma_l R_l \cdot N(0,1) \right] \tag{7}$$

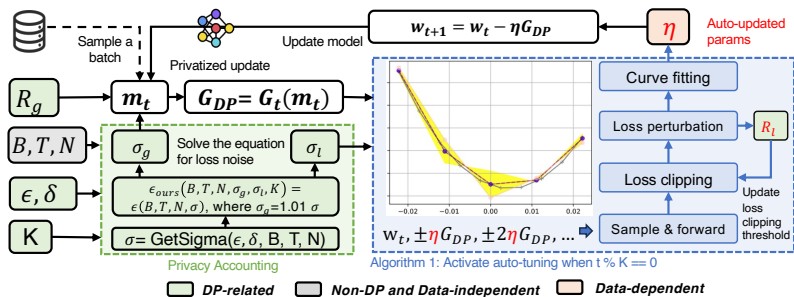

Figure 1: HyFreeDP overview with three types of hyper-parameters in the DP training. HyFreeDP saves tuning efforts via automatically tuning hyper-parameters in red text, and sets other parameters as default constants. We showcase with 5 points in curve fitting.

We note (7) is a private mean estimation that has been heavily studied in previous works (Biswas et al., 2020; Kamath et al., 2020), though many use an asymptotic threshold like $R_l + O(\log B)$, whereas Theorem 1 is more suited for our application in practice.

### 3.2 BIAS-VARIANCE TRADE-OFF IN LOSS PRIVATIZATION

**Theorem 1.** *The per-sample clipping bias of* (7) *is*

$$\left| \mathbb{E}(\tilde{L}) - \frac{\sum_i L_i}{B} \right| = \left| \frac{1}{B} \sum_i \min(\frac{R_l}{L_i}, 1)L_i - \frac{1}{B} \sum_i L_i \right| = \left| \frac{1}{B} \sum_i (L_i - R_l)\mathbb{I}(L_i > R_l) \right|$$

*which is monotonically decreasing in $R_l$, and converges to $[\mathbb{E}(L_i | L_i > R_l) - R_l] \cdot \mathbb{P}(L_i > R_l)$ as $B \to \infty$. In contrast, the noise variance is $\mathrm{Var}(\tilde{L}) = (\sigma_l R_l / B)^2$ which is increasing in $R_l$.*

In words, a large $R_l$ reduces the clipping bias but magnifies the noise, and vice versa for a small $R_l$. We propose to use $R_l \approx L$ so that the clipping bias is close to zero (i.e. $\tilde{L}$ is approximately unbiased), and the loss noise shown in Table 1 is reasonably small for large batch size.

To put this into perspective, we give the explicit form of clipping bias when $L_i$ follows a Gaussian distribution in Corollary 1.

**Corollary 1.** *Suppose $L_i \sim N(\mu, \xi^2)$, then the asymptotic clipping bias in Theorem 1 is*

$$[\mathbb{E}(L_i | L_i > R_l) - R_l] \cdot \mathbb{P}(L_i > R_l) = \xi[\phi(\alpha) - \alpha(1 - \Phi(\alpha)], \tag{8}$$

*where $\alpha = \frac{R_l - \mu}{\xi}$, $\phi$ is the probability density function and $\Phi$ is the cumulative distribution function of standard normal distribution. The term* (8) *is strictly decreasing in $\alpha$ as well as $R_l$ (see Figure 7).*

## 4 ALGORITHM

### 4.1 HYPERPARAMETER-FREE DP OPTIMIZATION

We present our algorithm in Algorithm 1, which is DP as guaranteed in Theorem 2, almost as efficient as the standard non-DP optimization by Section 4.4, and highly accurate and fast in convergence as demonstrated in Section 5. Importantly, we have split the hyperparameters into three classes, as shown in Figure 1:

- **DP-related** hyperparameters that do not depend on the tasks, such as the gradient noise $\sigma_g$, the loss noise $\sigma_l$, and the update interval $K$, can be set as default constants For example, we fix $R_g \to 0^+$ and re-scale the learning rate by $1/R_g$ according to automatic clipping and set $K = 5$.

- Training hyperparameters that are robust to different models and datasets, which we view as **data-independent**, need-not-to-search, and not violating DP, such as the batch size $B$ and the number of iterations $T$. We also fix other hyperparameters not explicitly displayed

in Algorithm 1, e.g. throughout this paper, we fix the momentum coefficients and weight decay $(\beta_1, \beta_2, \text{weight\_decay}) = (0.9, 0.999, 0.01)$, which is the default in Pytorch AdamW.

- Training hyperparameters that are **data-dependent**, which requires dynamical searching under DP, such as $R_l$ and $\eta$. For these hyperparameters, Algorithm 1 adopts multiple *auto-regressive* designs, i.e. the variables to use in the $t$-th iteration is based on the $(t-1)$-th iteration, which has already been privatized. These auto-regressive designs allow new variables to preserve DP by the post-processing property of DP[1].

To be specific, we have used $\tilde{L}_{t-1}^{(0)}$ as the loss clipping threshold $R_l$ for the next iteration in Line 7, because loss values remain similar values within a few iterations; In practice, we set a more conservative loss clipping threshold $R_l = \sum \tilde{L}_{t-1}^{(k)}$ to avoid the clipping bias. We have used the previous $\eta$ to construct next-loss in Line 6, which in turn will determine the new $\eta$ by Line 9.

---

**Algorithm 1** Hyperparameter-free Optimization with Differential Privacy

1: **INPUT:** initial $\eta$=1e-4, initial $R_l = 1$
2: Forward pass to compute per-sample losses $L_{t,i}^{(0)} = L(\boldsymbol{w}_t, \boldsymbol{x}_i)$
3: Compute the mini-batch loss $L_t^{(0)} = \frac{1}{B} \sum_i L_{t,i}^{(0)}$
4: Back-propagate from $L_t^{(0)}$ to compute $\boldsymbol{m}_{\text{DP}}$ in (2) with automatic clipping
5: Post-process $\boldsymbol{m}_{\text{DP}}$ by any optimizer $\mathbf{G}_{\text{DP}} := \mathbf{G}(\boldsymbol{m})$ in (3)
6: **if** $t\%K == 0$ (e.g. $K = 10$) **then**
7:     Forward pass to get per-sample losses $L_{t,i}^{(\pm 1)} = L(\boldsymbol{w}_t \pm \eta \mathbf{G}_{\text{DP}}, \boldsymbol{x}_i)$
8:     Privatize losses $\tilde{L}_t^{(k)}$ by (7) with $R_l = \tilde{L}_{t-1}^{(0)}$ for $k \in \{-1, 0, +1\}$
9:     Fit the quadratic function in (6) from $\{-\eta, 0, \eta\}$ to $\{\tilde{L}_t^{(-1)}, \tilde{L}_t^{(0)}, \tilde{L}_t^{(+1)}\}$
10:     Extract coefficients of the fitted quadratic function $(\boldsymbol{m}^\top \mathbf{H} \boldsymbol{m})_{\text{DP}}, (\mathbf{G}^\top \boldsymbol{m})_{\text{DP}}$
11:     Update $\eta$ with $\eta_{\text{GeN-DP}} = \frac{(\mathbf{G}^\top \boldsymbol{m})_{\text{DP}}}{(\boldsymbol{m}^\top \mathbf{H} \boldsymbol{m})_{\text{DP}}} \approx \arg\min_\eta \tilde{L}(\boldsymbol{w}_t - \eta \mathbf{G}_{\text{DP}})$
12: **end if**
13: Update $\boldsymbol{w}_{t+1} = \boldsymbol{w}_t - \eta \mathbf{G}_{\text{DP}}$

---

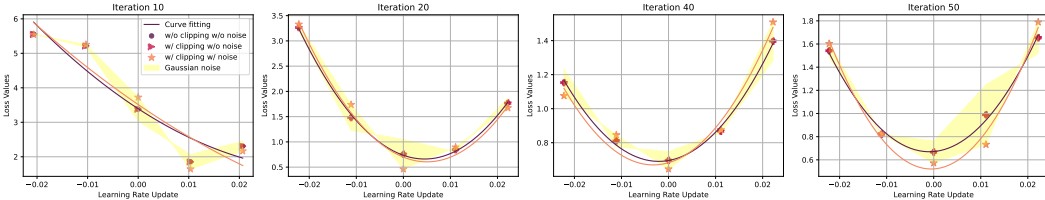

Figure 2: Impact of loss value clipping and perturbation on curve fitting along different training iterations on CIFAR100 with Vit-Small fully fine-tuning, with zero in x-axis denotes the current $\boldsymbol{w}_t$. We use 5 points for the ease of illustration and use 3 points in Algorithm 1 and experiments.

## 4.2 PRIVACY GUARANTEE

**Theorem 2.** *Algorithm 1 is $(\epsilon_{ours}, \delta)$-DP, where $\epsilon_{ours}$ depends on the batch size $B$, the number of iterations $T$, the noises $(\sigma_g, \sigma_l)$, and the update interval $K$. In contrast, vanilla DP-SGD is $(\epsilon_{vanilla}, \delta)$-DP, where $\epsilon_{vanilla}$ depends on $B, T, \sigma$. Furthermore, we have $\epsilon_{ours}(B, T, N, \sigma_g, \sigma_l, K) > \epsilon_{vanilla}(B, T, N, \sigma)$ if $\sigma_g = \sigma$.*

We omit the concrete formulae of $\epsilon$ because it depends on the choice of privacy accountants. For example, if we use $\mu$-GDP as an asymptotic estimation, then we show in Appendix B that

$$\mu_{\text{vanilla}} = \sqrt{\left(\frac{B}{N}\right)^2 T(e^{1/\sigma^2} - 1)}, \mu_{\text{ours}} = \sqrt{\mu_{\text{vanilla}}^2 + \left(\frac{B}{N}\right)^2 \frac{3T}{K}(e^{1/\sigma_l^2} - 1)} \quad (9)$$

---

[1]The post-processing of DP ensures that if $X$ is $(\epsilon, \delta)$-DP then $g(X)$ is also $(\epsilon, \delta)$-DP for any function $g$.

which can translate into $(\epsilon, \delta)$-DP by Equation (6) in Bu et al. (2020). Note in our experiments, we use the improved RDP as the privacy accountant.

Theorem 2 and (9) show that given the same $(B, T, \sigma)$, our Algorithm 1 uses more privacy budget because we additionally privatize the loss, whereas the vanilla DP-SGD does not protect the hyper-parameter tuning. To maintain the same privacy budget as vanilla DP-SGD, we use $\approx 99\%$ budget for the gradient privatization and $\approx 1\%$ budget for the loss privatization. Therefore, we need $\approx 1\%$ larger gradient noise $\sigma_g = \gamma \sigma$ and then select $\sigma_l$ based on

$$\epsilon_{\text{ours}}(B, T, N, \gamma\sigma, \sigma_l, K) = \epsilon_{\text{vanilla}}(B, T, N, \sigma), \text{where we can use } \gamma \leq 1.01. \quad (10)$$

### 4.3 END-TO-END NOISE DETERMINATION

We use `autoDP` library[2] to compute $\sigma_l$ based on $\sigma_g$ in (10). We give more details of our implementation in Appendix C. We visualize both noises in Figure 3 with RDP accounting and Gaussian and mechanisms for loss privatization, dashed line indicates the case when $\sigma_g$ and $\sigma_l$ are set equally. More examples for different mechanisms or different accounting are shown in Appendix. Note that we only add a little more gradient noise, hence $\sigma_g$ introduces negligible accuracy drop to Algorithm 1, as empirically shown in Section 5. Additionally, we demonstrate in Figure 2 that $\sigma_l$ has negligible interference with the precision of estimating $\eta_{\text{GeN}}$.

### 4.4 EFFICIENCY OF ALGORITHM

We illustrate that Algorithm 1 can be almost as efficient as the standard non-DP optimization, in terms of training time and memory cost. We identify three orthogonal components that are absent from non-DP optimization: **(1) Gradient privatization.** DP optimization (including vanilla DP-SGD) always requires per-sample gradient clipping. Due to the high dimension of gradients, this could incur high cost in memory and time if implemented inefficiently. We directly leverage the recent advances like ghost clipping and book-keeping (BK) which have allowed DP optimization to be almost as efficient as non-DP optimization, up to 256 GPUs and 100B parameters. **(2) Loss privatization.** The cost of loss privatization alone is $O(B)$ and thus negligible, compared to the forward

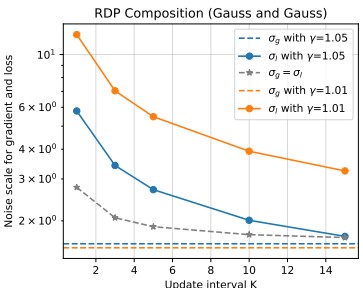

Figure 3: Gradient and loss noise.

passes and back-propagation which are $O(Bd)$. **(3) Learning rate computation.** The cost of computing $\eta_{\text{GeN-DP}}$ in (6) mainly comes from the additional forward passes[3] for $L_{t,i}^{(\pm 1)}$. Given that the back-propagation approximately costs $2\times$ the training time of forward pass, the optimizer without GeN learning rate (non-DP or DP) roughly uses 3 units of time at each iteration. In contrast, Algorithm 1 uses $3 + 2/K \approx 3.2$ units if we set $K = 10$ with $< 7\%$ overhead. We emphasize that the actual overhead to training time is even lower, because the training also includes non-optimization operations such as data loading and inter-GPU communication. In short, the efficiency gap between Algorithm 1 and non-DP optimization is negligible in practice.

## 5 EXPERIMENTS

To comprehensively evaluate the effectiveness of the proposed method, we conduct experiments on both computer vision tasks and natural language tasks, across different model architectures (Vit (Yuan et al., 2021), GPT2 (Radford et al., 2019) and LLaMa2-7B (Touvron et al., 2023)) and fine-tuning paradigms (Full, BitFit (Zaken et al., 2022) and LoRA (Hu et al., 2021)). BitFit only tunes the bias terms of a pre-trained model, while LoRA only tunes the injected low-rank matrices, both keeping all other parameters frozen. We use the privacy budget $\epsilon = \{1, 3, 8\}$ with $\delta \ll N^{-1.1}$ for training dataset with $N$ samples[4] and also perform non-DP baseline ($\epsilon = \infty$).

---

[2]https://github.com/yuxiangw/autodp

[3]The number of $\{\eta_i\}_i$ in (6) is at least two since there are two unknown variables. More $\eta_i$ may stabilize the algorithm, at cost of more forward passes, longer training time, and using more privacy budget.

[4]https://github.com/lxuechen/private-transformers

Table 2: Performance comparison of HyFreeDP with other baselines. We use cosine learning rate decay for NonDP-GS w/ LS in the BitFit fine-tuning. Detailed results are provided in the Appendix.

| Fully Fine-Tune | | Vit-Small | | | | | Vit-Base | | | | |
|---|---|---|---|---|---|---|---|---|---|---|---|
| Privacy budget | Method | CIFAR10 | CIFAR100 | SVHN | GTSRB | Food101 | CIFAR10 | CIFAR100 | SVHN | GTSRB | Food101 |
| $\epsilon = 1$ | NonDP-GS | 96.49 | 79.70 | 89.86 | 67.00 | 70.38 | 95.20 | 67.18 | 89.10 | 81.11 | 58.80 |
| | D-adaptation | 23.74 | 0.80 | 15.27 | 1.86 | 1.17 | 19.15 | 0.83 | 18.56 | 4.72 | 1.48 |
| | Prodigy | 27.54 | 0.80 | 15.27 | 1.86 | 1.19 | 19.15 | 0.83 | 18.56 | 4.72 | 1.43 |
| | DP-hyper | 92.98 | 74.63 | 34.58 | 28.23 | 19.06 | 94.86 | 6.59 | 79.94 | 77.85 | 57.02 |
| | HyFreeDP | 96.36 | 78.17 | 91.15 | 74.68 | 67.98 | 95.76 | 67.17 | 88.24 | 79.00 | 58.16 |
| $\epsilon = 3$ | NonDP-GS | 96.86 | 83.69 | 91.60 | 83.08 | 74.58 | 95.84 | 79.03 | 91.74 | 91.03 | 66.87 |
| | D-adaptation | 40.99 | 0.94 | 17.22 | 2.65 | 1.37 | 30.29 | 1.07 | 19.77 | 5.68 | 1.68 |
| | Prodigy | 77.18 | 0.95 | 18.11 | 2.68 | 1.47 | 30.29 | 1.07 | 19.77 | 5.68 | 1.65 |
| | DP-hyper | 95.10 | 78.82 | 48.82 | 42.02 | 36.21 | 95.75 | 19.92 | 87.82 | 90.25 | 66.09 |
| | HyFreeDP | 96.89 | 81.62 | 92.21 | 89.92 | 75.24 | 97.29 | 80.34 | 91.71 | 91.76 | 73.46 |
| NonDP | NonDP-GS | 98.33 | 89.41 | 95.58 | 98.66 | 85.19 | 98.88 | 92.40 | 96.87 | 98.41 | 89.61 |
| | D-adaptation | 54.29 | 86.35 | 97.09 | 97.38 | 74.75 | 63.37 | 88.41 | 97.02 | 98.65 | 78.35 |
| | Prodigy | 97.85 | 89.21 | 96.87 | 98.31 | 87.49 | 98.59 | 91.01 | 97.14 | 98.77 | 89.47 |
| | HyFreeDP | 98.32 | 90.84 | 96.86 | 99.00 | 86.38 | 98.83 | 92.58 | 96.75 | 97.37 | 88.73 |

| BitFit Fine-Tune | | Vit-Small | | | | | Vit-Base | | | | |
|---|---|---|---|---|---|---|---|---|---|---|---|
| Privacy budget | Method | CIFAR10 | CIFAR100 | SVHN | GTSRB | Food101 | CIFAR10 | CIFAR100 | SVHN | GTSRB | Food101 |
| $\epsilon = 1$ | NonDP-GS | 96.74 | 84.22 | 90.18 | 86.20 | 77.39 | 97.34 | 84.97 | 90.91 | 87.15 | 76.61 |
| | NonDP-GS w/ LS | 97.40 | 81.36 | 67.95 | 48.64 | 74.55 | 97.90 | 81.87 | 79.86 | 55.66 | 73.76 |
| | Prodigy | 96.36 | 82.23 | 90.30 | 86.85 | 76.36 | 97.13 | 84.87 | 91.08 | 81.54 | 78.72 |
| | HyFreeDP | 96.20 | 83.84 | 90.49 | 88.08 | 79.27 | 97.42 | 84.01 | 92.21 | 81.83 | 79.25 |
| $\epsilon = 3$ | NonDP-GS | 97.13 | 85.95 | 91.42 | 91.50 | 80.37 | 97.73 | 87.00 | 92.20 | 91.46 | 80.06 |
| | NonDP-GS w/ LS | 97.61 | 84.87 | 78.91 | 60.13 | 78.08 | 98.05 | 85.48 | 86.22 | 69.03 | 78.71 |
| | Prodigy | 95.72 | 83.63 | 91.90 | 91.64 | 79.20 | 96.96 | 86.26 | 92.13 | 90.27 | 81.98 |
| | HyFreeDP | 97.09 | 86.07 | 91.64 | 92.38 | 84.55 | 97.65 | 87.04 | 92.37 | 90.67 | 84.53 |
| NonDP | NonDP-GS | 97.91 | 89.39 | 91.88 | 95.20 | 86.29 | 98.56 | 91.59 | 94.42 | 92.87 | 88.37 |
| | NonDP-GS w/ LS | 97.89 | 89.40 | 91.98 | 90.31 | 86.27 | 98.56 | 91.61 | 94.43 | 92.90 | 88.39 |
| | Prodigy | 97.88 | 87.85 | 95.19 | 95.34 | 86.56 | 98.41 | 90.70 | 96.02 | 95.02 | 88.60 |
| | HyFreeDP | 97.97 | 89.86 | 93.60 | 95.19 | 87.24 | 98.43 | 91.69 | 95.48 | 95.23 | 89.06 |

As the first hyperparameter-free method for differentially private optimization, we compare HyFreeDP with the following baselines: 1) **NonDP-GS**: We manually perform grid search over a predefined range of learning rates, selecting the best without accumulating privacy budget across runs. This serves as the performance upper bound since tuning is non-DP. We also experiment with a manually tuned learning rate scheduler, noted as NonDP-GS w/ LS. We search the learning rate over the range [5e-5, 1e-4, 5e-4, 1e-3, 5e-3] based on previous works (Bu et al., 2023b;a) to cover suitable $\eta$ for various DP levels. 2) **DP-hyper** (Liu & Talwar, 2019; Wang et al., 2023; Papernot & Steinke, 2021): We simulate with a narrow range around the optimal $\eta$, spending 85% of the privacy budget for DP training with the searched $\eta$. 3) **D-Adaptation** (Defazio & Mishchenko, 2023) and **Prodigy** (Mishchenko & Defazio, 2024): Both are state-of-the-art learning rate tuning algorithms in non-DP optimization. We adopt their optimizers with recommended hyperparameters, alongside DP-specific clipping and perturbation. 4) **HyFreeDP** : We initialize $R_l = 1$ and $\eta = 1e-4$ by default, allowing automatic updates in training.

## 5.1 IMAGE CLASSIFICATION TASKS

**Experimental setups.** As the main result shown in Table 2, we compare HyFreeDP to other baselines by experiments on CIFAR10, CIFAR100, SVHN, GTSRB and Food101 for models of Vit-Small and Vit-Base.

**Evaluation results.** HyFreeDP almost outperforms all end-to-end DP baselines. While non-DP automatic learning rate schedulers (D-adaptation and Prodigy) sometimes match grid-searched constants, they perform poorly in DP training, especially with tighter privacy budgets and when the number of trainable parameters is large, confirming our analysis in Section 2.3. Although DP-hyper surpasses these schedulers—likely due to our intentionally narrow search range—finding suitable ranges remains challenging as training dynamics vary by dataset and privacy budget. Even in this optimistic setting, DP-hyper underperforms due to increased gradient noise. In contrast, HyFreeDP achieves consistent performance across various $\epsilon$ values and datasets without manual learning rate tuning, thanks to our gradient noise control and efficient learning rate estimation with minimal loss value perturbation.

HyFreeDP achieves comparable or superior performance to NonDP-GS baseline without manual learning rate tuning in BitFit experiments. NonDP-GS w/ LS and Prodigy show improved stability compared to full fine-tuning, suggesting the sensitivity to training paradigms and trainable model size. In Figure 4, we illustrate training dynamics ($R_l$, $\eta$, loss, test accuracy) across meth-

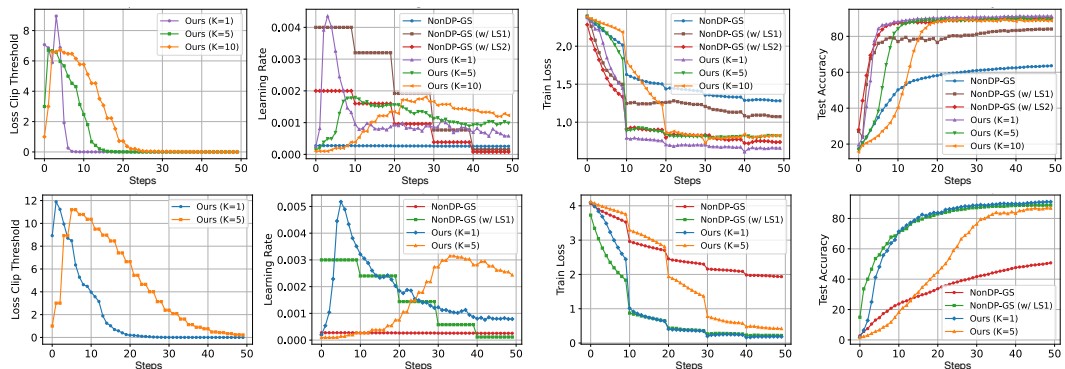

Figure 4: Automatic learning of clipping threshold, learning rate, training loss, and testing accuracy for SVHN (top) and GTSRB (bottom). HyFreeDP schedules $R_l$ and $\eta$ during training, approaching the manually tuned baseline with end-to-end DP guarantees, and is robust to varying intervals $K$.

ods. HyFreeDP automatically finds optimal learning rate schedules, with clipping thresholds peaking early and decreasing gradually, enabling more accurate rate estimation as training progresses. While updating with $K = 1$ yields optimal convergence, HyFreeDP remains robust to less frequent updates (e.g., $K = 5$), balancing tuning cost and convergence speed.

## 5.2 NATURAL LANGUAGE GENERATION TASKS

**Experimental setups.** We conduct experiments on E2E dataset with a table to text task on GPT-2 model, and also evaluate the language generation task with PubMed dataset by fine-tuning LLaMa2-7B model with LoRA (Hu et al., 2021) for demonstrating the scalability and generality of HyFreeDP. We follow the experimental setups based on previous works (Bu et al., 2024a) and use the dataset provided by Yu et al. (2022; 2023) which contains over 75,000 abstracts of medical papers that were published after the cut-off date of LLaMa2. Based on the non-DP experience, LoRA typically requires a magnitude greater learning rate than full fine-tuning[5], thus we scale up our default initial learning by $\times 10$. We tune the best learning rate for LLaMa2-7B with LoRA fine-tuning on 4,000 samples of PubMed for NonDP-GS when training on the full dataset.

Table 3: Performance comparison on GPT-2 for E2E dataset with different privacy budgets. Best end-to-end DP results are bolded, and results surpassing the manually tuned baseline are underlined.

| Full Fine-Tune | | $\epsilon = 3$ | | | | | $\epsilon = 8$ | | | | |
|---|---|---|---|---|---|---|---|---|---|---|---|
| Model | Method | BLEU | CIDEr | METEOR | NIST | ROUGE_L | BLEU | CIDEr | METEOR | NIST | ROUGE_L |
| GPT-2 | NonDP-GS | 0.583 | 1.566 | 0.367 | 5.656 | 0.653 | 0.612 | 1.764 | 0.385 | 6.772 | 0.664 |
| | D-Adaptation | 0.000 | 0.000 | 0.003 | 0.082 | 0.016 | 0.000 | 0.000 | 0.000 | 0.000 | 0.000 |
| | Prodigy | 0.082 | 0.000 | 0.157 | 1.307 | 0.239 | 0.012 | 0.000 | 0.003 | 0.000 | 0.003 |
| | HyFreeDP | **0.585** | **1.564** | **0.365** | **5.736** | **0.636** | **0.612** | **1.768** | **0.378** | **6.702** | **0.655** |

**Evaluation results.** As shown in Table 3, we observe that even when the privacy budget is not small (e.g., $\epsilon = 8$), non-DP automatic learning rate scheduler does not perform well. We find that HyFreeDP consistently obtains a comparable performance as the NonDP-GS baseline without extra tuning. In Figure 5, we observe that HyFreeDP automatically discovers a learning rate schedule that achieves better generalization performance compared to the early-stopped NonDP-GS. The automatically determined learning rate ($\eta$) reveals a consistent pattern across model scales: an

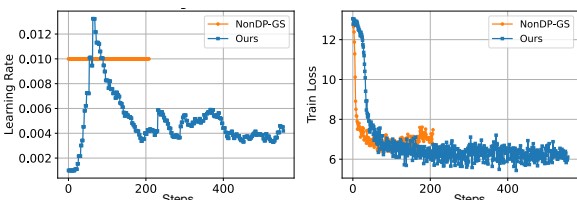

Figure 5: Training dynamics of Llama2-7B on PubMed

---

[5]See this reference as an example. Note that LoRA may use a similar learning rate if it is set proportional to rank (Biderman et al., 2024).

"increase-then-decrease" trajectory that holds true from smaller models (Vit-Small) to larger models (LLaMa2-7B).

Table 4: Comparison of model performance in minutes with and without HyFreeDP across various datasets. The coefficients represent the ratio relative to the w/o auto configuration.

| Models | Dataset | K=1 | K=5 | K=10 | w/o HyFreeDP |
|---|---|---|---|---|---|
| Llama2-7B (LoRA-FT) | PubMed (4k) | 409.750 ($\times$ 2.040) | 244.333 ($\times$ 1.217) | 222.583 ($\times$ 1.108) | 200.833 ($\times$ 1.000) |
| GPT2 | E2E | 163.333 ($\times$ 1.888) | 97.167 ($\times$ 1.123) | 94.983 ($\times$ 1.098) | 86.500 ($\times$ 1.000) |
| Vit-base | CIFAR100 | 152.617 ($\times$ 1.370) | 118.483 ($\times$ 1.063) | 113.317 ($\times$ 1.017) | 111.433 ($\times$ 1.000) |
| Vit-base (BitFit-FT) | CIFAR100 | 113.450 ($\times$ 1.654) | 74.733 ($\times$ 1.089) | 73.817 ($\times$ 1.076) | 68.600 ($\times$ 1.000) |
| Vit-small | SVHN | 102.000 ($\times$ 1.255) | 84.500 ($\times$ 1.040) | 82.800 ($\times$ 1.019) | 81.250 ($\times$ 1.000) |

## 5.3 EFFICIENCY COMPARISON

Based on Section 4.4, we compare the training efficiency of HyFreeDP with a single run of DP training using the same non-DP and data-independent hyperparameters, as shown in Table 4. For LLaMa2-7B, we sample 4,000 records from PubMed and train for 3 epochs on a single A100 (80GB). For smaller datasets and models, we use previous setups on Titan RTX (24GB). HyFreeDP introduces less than $\times 2$ overhead compared to a single run of DP training, even with frequent updates at $K = 1$. Smaller models or those using LoRA or BitFit have lower additional costs, especially with $K = 1$, and the gap narrows as $K$ increases, approaching a cost factor of $1\times$.

## 6 DISCUSSION AND CONCLUSION

In conclusion, we tackle the challenge of hyperparameter tuning in differential privacy (DP) by introducing a hyperparameter-free DP training method that privately and automatically updates the learning rate. Combined with automatic clipping, our approach reduces tuning efforts and ensures end-to-end DP during training. This bridges the gap between hyperparameter-free methods in non-DP settings and DP optimization, opening promising avenues for future research.

## ACKNOWLEDGMENTS

R.L. is partially supported by National Institutes of Health grant (R01LM013712), National Science Foundation grants (CNS-2124104 and CNS-2125530). We sincerely thank Prof. Li Xiong for her invaluable support and guidance throughout this work. We also appreciate the insightful comments and constructive feedback from the reviewers and the area chair.

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

## A EXPERIMENTAL DETAILS

**Image classification.** In full fine-tuning, we tried to integrate a constant linear scheduler as a common practice, but the result does not steadily outperform the tuned NonDP-GS, so we omit the results in Appendix Table 5. We do not apply the linear scheduler to Prodigy and D-adaptation for keeping the originally recommended configuration. The results with $\epsilon = 8$ are shown in Table 7 with the consistent conclusions across different datasets.

| | Vit-Small | | | | | Vit-Base | | | | |
|---|---|---|---|---|---|---|---|---|---|---|
| Privacy Budget | CIFAR10 | CIFAR100 | SVHN | GTSRB | Food101 | CIFAR10 | CIFAR100 | SVHN | GTSRB | Food101 |
| $\epsilon = 1$ | 88.02 | 3.46 | 29.00 | 10.16 | 9.80 | 96.62 | 58.74 | 89.13 | 64.39 | 60.74 |
| $\epsilon = 3$ | 92.48 | 8.00 | 35.54 | 17.68 | 20.40 | 97.11 | 75.73 | 91.79 | 82.31 | 69.09 |
| $\epsilon = 8$ | 93.79 | 15.04 | 43.73 | 24.15 | 31.05 | 97.41 | 81.31 | 93.05 | 88.79 | 73.65 |
| NonDP | 98.00 | 89.11 | 96.55 | 96.94 | 84.55 | 98.88 | 92.51 | 97.27 | 98.29 | 89.10 |

Table 5: Full fine-tuning results by adding a linear scheduler to NonDP-GS across different datasets with privacy budgets for Vit-Small and Vit-Base models. Results show that directly integrating a constant learning rate scheduler in NonDP does not hurt performance but the DP training performance is sensitive to the learning rate scheduler.

In Table 2, we report the test accuracy averaged over 3 checkpoints. For demonstrating the stability of different methods, we present the mean accuracy with standard deviation in Table 6 for low privacy budget regime and NonDP.

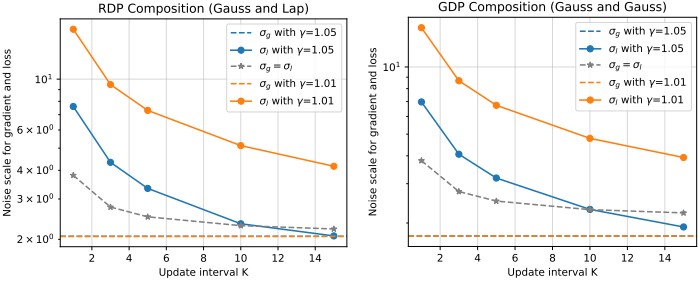

Figure 6: Privacy composition for gradient and loss values privatization, with privacy accountants of RDP and GDP, and loss perturbation of Gaussian and Laplacian mechanisms. Gray dashed line indicates the naive and even budget splitting for every access to private gradient and loss, which results in larger noise magnitude on gradient especially when the adjustment is frequent. The privacy accounting strategy proposed in HyFreeDP effectively restrains the privacy budget consumption on hyper-parameter tuning and spending it wisely by only perturbing a single-dimensional loss value.

**Clipping bias.** Additionally, we demonstrate the loss clipping bias in Figure 7.

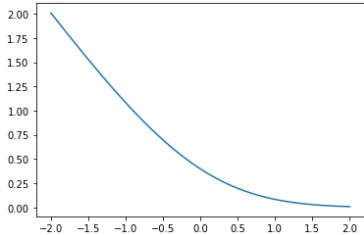

Figure 7: $\phi(\alpha) - \alpha(1 - \Phi(\alpha))$ which is minimized at $\alpha = \infty$.

Table 6: Averaged test accuracy with standard deviation of HyFreeDP with other baselines

| Fully Fine-Tune | | Vit-Small | | | | | Vit-Base | | | | |
|---|---|---|---|---|---|---|---|---|---|---|---|
| Privacy budget | Method | CIFAR10 | CIFAR100 | SVHN | GTSRB | Food101 | CIFAR10 | CIFAR100 | SVHN | GTSRB | Food101 |
| $\epsilon=1$ | NonDP-GS | 96.49 ±0.02 | 79.7 ±0.13 | 89.86 ±0.21 | 67.00 ±0.44 | 70.38 ±0.23 | 95.2 ±0.02 | 67.18 ±0.52 | 89.1 ±0.08 | 81.11 ±0.27 | 58.8 ±0.32 |
| | D-adaptation | 23.74 ±0.19 | 0.8 ±0.01 | 15.27 ±0.08 | 1.86 ±0 | 1.17 ±0.01 | 19.15 ±0.41 | 0.83 ±0.31 | 18.56 ±0.74 | 4.72 ±0.02 | 1.48 ±0.10 |
| | Prodigy | 27.54 ±0.61 | 0.80 ±0.01 | 15.27 ±0.08 | 1.86 ±0.00 | 1.19 ±0.00 | 19.15 ±0.41 | 0.83 ±0.00 | 18.56 ±0.04 | 4.72 ±0.02 | 1.43 ±0.00 |
| | DP-hyper | 92.98 ±0.14 | 74.63 ±0.09 | 34.58 ±0.58 | 28.23 ±0.91 | 19.06 ±0.59 | 94.86 ±0.10 | 6.59 ±0.00 | 79.94 ±0.04 | 77.85 ±0.16 | 57.02 ±0.00 |
| | HyFreeDP | 96.36 ±0.03 | 78.17 ±0.12 | 91.15 ±0.06 | 74.68 ±0.54 | 67.98 ±0.09 | 95.76 ±0.03 | 67.17 ±0.11 | 88.24 ±0.14 | 79.00 ±0.51 | 58.16 ±0.07 |
| $\epsilon=3$ | NonDP-GS | 96.86 ±0.03 | 83.69 ±0.12 | 91.6 ±0.11 | 83.08 ±0.69 | 74.58 ±0.23 | 95.84 ±0.07 | 79.03 ±0.18 | 91.74 ±0.04 | 91.03 ±0.11 | 66.87 ±0.06 |
| | D-adaptation | 40.99 ±0.92 | 0.94 ±0.01 | 17.22 ±0.09 | 2.65 ±0.03 | 1.37 ±0.01 | 30.29 ±0.51 | 1.07 ±0.01 | 19.77 ±0.04 | 5.68 ±0.00 | 1.68 ±0.01 |
| | Prodigy | 77.18 ±1.85 | 0.95 ±0.01 | 18.11 ±0.16 | 2.68 ±0.05 | 1.47 ±0.00 | 30.29 ±0.51 | 1.07 ±0.01 | 19.77 ±0.04 | 5.68 ±0.00 | 1.65 ±0.02 |
| | DP-hyper | 95.10 ±0.04 | 78.82 ±0.14 | 48.82 ±1.15 | 42.02 ±0.47 | 36.21 ±0.8 | 95.75 ±0.09 | 19.92 ±0.73 | 87.82 ±0.19 | 90.25 ±0.42 | 66.09 ±0.05 |
| | HyFreeDP | 96.89 ±0.06 | 81.62 ±0.06 | 92.21 ±0.12 | 89.92 ±0.57 | 75.24 ±0.06 | 97.29 ±0.02 | 80.34 ±0.05 | 91.71 ±0.18 | 91.76 ±0.48 | 73.46 ±0.06 |
| NonDP | NonDP-GS | 98.33 ±0.01 | 89.41 ±0.18 | 95.58 ±0.14 | 98.66 ±0.02 | 85.19 ±0.12 | 98.88 ±0.01 | 92.4 ±0.04 | 96.87 ±0.11 | 98.41 ±0.02 | 89.61 ±0.08 |
| | D-adaptation | 54.29 ±0.53 | 86.35 ±0.06 | 97.09 ±0.08 | 97.38 ±0.31 | 74.75 ±1.00 | 63.37 ±1.10 | 88.41 ±0.05 | 97.02 ±0.09 | 98.65 ±0.01 | 78.35 ±1.17 |
| | Prodigy | 97.85 ±0.03 | 89.21 ±0.07 | 96.87 ±0.07 | 98.31 ±0.14 | 87.49 ±0.08 | 98.59 ±0.06 | 91.01 ±0.10 | 97.14 ±0.06 | 98.77 ±0.14 | 89.47 ±0.06 |
| | HyFreeDP | 98.32 ±0.03 | 90.84 ±0.02 | 96.86 ±0.02 | 99.00 ±0.02 | 86.38 ±0.03 | 98.83 ±0.01 | 92.58 ±0.02 | 96.75 ±0.05 | 97.37 ±0.05 | 88.73 ±0.02 |

Table 7: Performance comparison of HyFreeDP to other baselines. We use consine learning rate decay for NonDP-GS w/ LS baseline in the BitFit fine-tuning setting.

| Full Fine-Tune | | Vit-Small | | | | | Vit-Base | | | | |
|---|---|---|---|---|---|---|---|---|---|---|---|
| Privacy budget | Method | CIFAR10 | CIFAR100 | SVHN | GTSRB | Food101 | CIFAR10 | CIFAR100 | SVHN | GTSRB | Food101 |
| $\epsilon=8$ | NonDP-GS | 96.28 | 84.99 | 92.53 | 89.97 | 77.08 | 96.14 | 82.52 | 92.38 | 94.91 | 71.05 |
| | D-adaptation | 78.31 | 1.07 | 19.10 | 3.67 | 1.76 | 40.90 | 1.25 | 20.86 | 6.84 | 1.94 |
| | Prodigy | 95.74 | 1.29 | 20.75 | 4.86 | 2.92 | 45.89 | 1.25 | 20.86 | 6.84 | 1.90 |
| | DP-hyper | 95.70 | 80.58 | 64.72 | 50.10 | 49.18 | 96.28 | 36.46 | 90.27 | 94.62 | 70.63 |
| | HyFreeDP | 97.04 | 86.00 | 95.15 | 88.06 | 80.61 | 97.79 | 87.57 | 95.00 | 94.07 | 81.91 |
| BitFit Fine-Tune | | Vit-Small | | | | | Vit-Base | | | | |
| Privacy budget | Method | CIFAR10 | CIFAR100 | SVHN | GTSRB | Food101 | CIFAR10 | CIFAR100 | SVHN | GTSRB | Food101 |
| $\epsilon=8$ | NonDP-GS | 97.23 | 86.56 | 92.23 | 93.34 | 82.12 | 97.81 | 88.05 | 93.38 | 93.04 | 81.88 |
| | NonDP-GS w/ LS | 97.63 | 86.21 | 83.17 | 67.35 | 79.85 | 98.08 | 87.17 | 88.25 | 75.61 | 81.18 |
| | HyFreeDP | 97.31 | 88.43 | 89.17 | 93.54 | 74.61 | 97.90 | 90.34 | 92.36 | 93.14 | 87.14 |

# B PROOFS

*Proof of Theorem 1.* It's not hard to see

$$\mathbb{E}(\tilde{L}) - \frac{\sum_i L_i}{B} = \frac{1}{B}\sum_i \min(\frac{R_l}{L_i}, 1)L_i - \frac{1}{B}\sum_i L_i$$

$$= \frac{1}{B}\sum_i (R_l - L_i)\mathbb{I}(L_i > R_l) = -\frac{1}{B}\sum_i \text{ReLU}(L_i - R_l)$$

in which the non-negative $\text{ReLU}(x) = x \cdot \mathbb{I}(x > 0)$. Taking the absolute value, we have

$$\left|\mathbb{E}(\tilde{L}) - \frac{\sum_i L_i}{B}\right| = \frac{1}{B}\sum_i \text{ReLU}(L_i - R_l) = \left|\frac{1}{B}\sum_i (L_i - R_l)\mathbb{I}(L_i > R_l)\right|$$

which is decreasing in $R_l$ because ReLU is increasing in its input $(L_i - R_l)$, and this input is decreasing in $R_l$.

As $B \to \infty$, the clipping bias tends to

$$\mathbb{E}(\text{ReLU}(L_i - R_l)) = \mathbb{E}(\text{ReLU}(L_i - R_l)|L_i > R_l) \cdot \mathbb{P}(L_i > R_l)$$

$$= \mathbb{E}((L_i - R_l)|L_i > R_l) \cdot \mathbb{P}(L_i > R_l) = [\mathbb{E}(L_i|L_i > R_l) - R_l] \cdot \mathbb{P}(L_i > R_l)$$

where we have used $\text{ReLU}(x) = x$ when $x > 0$. □

*Proof of Corollary 1.* The key part in (8) is $\mathbb{E}(L_i|L_i > R_l)$, which is the expectation of the truncated normal distribution by one-side truncation. It is known that for $\alpha = \frac{R_l - \mu}{\xi}$,

$$\mathbb{E}(L_i|L_i > R_l) = \mu + \xi\frac{\phi(\alpha)}{1 - \Phi(\alpha)}, \quad \mathbb{P}(L_i > R_l) = 1 - \Phi(\alpha)$$

The proof is complete by inserting these quantities. □

*Proof of Theorem 2.* All privacy budget of Algorithm 1 goes into two components: privatizing the gradient (with noise level $\sigma_g$) and privatizing the loss (with noise level $\sigma_l$).

Under the same $(B, T, N, \sigma_g)$, we have $T$ mechanisms of gradient privatization, each of $(\epsilon_g, \delta_g)$-DP and $3T/K$ mechanisms of loss privatization, each of $(\epsilon_l, \delta_l)$-DP. Hence it is clear that $\epsilon_{\text{ours}} > \epsilon_{\text{vanilla}}$.

To be more specific, we demonstrate with $\mu$-GDP. The vanilla DP-SGD is $\mu$-GDP with

$$\mu_{\text{vanilla}} = \frac{B}{N} \sqrt{T(e^{1/\sigma_g^2} - 1)}$$

which is the same as the gradient privatization component of our DP-SGD. We additionally spend

$$\mu_l = \frac{B}{N} \sqrt{\frac{3T}{K}(e^{1/\sigma_l^2} - 1)}$$

leading to a total budget of

$$\mu_{\text{ours}} = \sqrt{\mu_{\text{vanilla}}^2 + \mu_l^2}$$

by Corollary 3.3 in Dong et al. (2022). It is clear $\mu_{\text{ours}} > \mu_{\text{vanilla}}$. $\qquad\square$

## C   END-TO-END PRIVACY ACCOUNTING AND INVERSE

We demonstrate how to determine $(\sigma_g, \sigma_l)$ given $(\epsilon, \delta)$-DP budget. In vanilla DP optimization, we can leverage privacy accountants such as RDP, GDP, PRV, etc. Each accountant is a function whose input is hyperparameters $(B, T, N, \delta, \sigma)$ and the output is $\epsilon$ (see an example in (9)). In this section, we denote any accountant as $f$, so that

$$f(\sigma; B, N, \delta) = \epsilon' \tag{11}$$

$$f^T(\sigma; B, N, \delta) = \epsilon \tag{12}$$

$$f^{-T}(\epsilon; B, N, \delta) = \sigma \tag{13}$$

in which $\epsilon'$ is the single-iteration budget, $f^T$ means a composition of $T$ iterations, and $f^{-T}$ is the inverse function known as `GetSigma` in Figure 1.

In this work, we develop an end-to-end privacy accountant to compose both the gradient privatization and the loss privatization. Our accountant takes the input $(B, T, N, \delta, \sigma_l, \gamma, K)$, where $\gamma = 1.01$ by default and can take smaller value for larger models or smaller $(\epsilon, \delta)$ budget. Therefore, $\sigma_g = \gamma\sigma$.

Firstly, we call $f^T(\gamma\sigma; B, N, \delta) = \hat{\epsilon}$ to get the reference budget $\hat{\epsilon}$ which is strictly smaller than $\epsilon$ because $f$ is monotonically decreasing in its input. Then we guess the loss noise and call $f^{3T/K}(\sigma_l; B, N, \delta) = \epsilon_l$ since there are $3T/K$ rounds of loss privatization. We continue our guess until

$$f^{3T/K}(\sigma_l; B, N, \delta) + f^T(\gamma\sigma; B, N, \delta) = \epsilon_l + \hat{\epsilon} = \epsilon$$

Notice the left hand side is monotonically decreasing in $\sigma_l$. Hence we use bisection method to find the unique solution $\sigma_l$, at an exponentially fast speed.

---

**Algorithm 2** End-to-End Privacy Accounting

---

1: **INPUT:** The end-to-end DP budget $(\epsilon, \delta)$, `GetSigma`$(\cdot)$, `Compose`$(\cdot)$, `Solve`$(\cdot)$
2: **OUTPUT:** Noise magnitude for gradient and loss privatization $\sigma_g$ and $\sigma_l$
3: ▷ *Compute the gradient noise scale $\sigma$ by assuming there is only a single training run*
4: $\sigma = $ `GetSigma`$(\epsilon, \delta, B, T, N)$
5: ▷ *Compute the $\sigma_g$ in Algorithm 1 with a controlled noise increase*
6: $\sigma_g = \gamma \cdot \sigma$ with the constant $\gamma$ slightly greater than 1 (e.g, $\gamma = 1.01$)
7: ▷ *Define the composition function with input variable as the loss noise scale $c$*
8: $\epsilon_{\text{ours}}(c|\sigma_g, \delta, B, N, T, K) = $ `Compose`$(f_g^T(\sigma_g; B, N, \delta), f_l^{3T/K}(c; B, N, \delta))$
9: ▷ *Solve the minimization of the scalar function respect to $c$*
10: $\sigma_l = $ `Solve`$(c, \epsilon) = \arg\min_c |\epsilon_{\text{ours}}(c) - \epsilon|$

---

In the above, we have used the functions `GetSigma`$(\cdot)$[6], `Compose`$(\cdot)$[7], and any root-finding method `Solve`$(\cdot)$ such as the bi-section in `scipy` library. We highlight that Algorithm 1 and

---

[6]`https://github.com/yuxiangw/autodp/blob/master/example/example_calibrator.py`

[7]`https://github.com/yuxiangw/autodp/blob/master/example/example_composition.py`

Algorithm 2 are sufficiently flexible to work with the general DP notions, including GDP, Renyi DP, tCDP, per-instance DP, per-user DP, etc.

# D   MISC

**Hyper-parameter matters for DP optimization.** Previous works De et al. (2022); Li et al. (2021) reveal that the performance of DP optimization is sensitive to the hyper-parameter choices, as we cited in Figure 8.

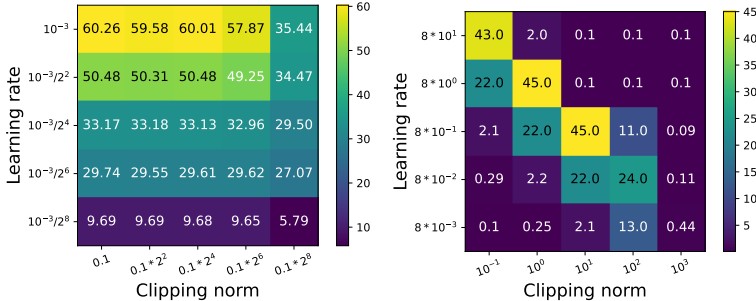

Figure 8: Hyperparameter tuning of $(R_g, \eta)$ cited from Figure 1 of Bu et al. (2023b). Here $R_g$ is the clipping norm and $\eta$ is the learning rate. Left: BLEU score of GPT2 on E2E dataset Li et al. (2021). Right: Test accuracy of ResNet18 on ImageNet Kurakin et al. (2022).

**End-to-end DP guarantee for optimization and tuning.** As shown in Table 8, we compare our work with representative works that try to ensure end-to-end DP for both optimization and tuning.

Table 8: Comparison of hyperparameter search strategies. Budget splitting percentages for Adaptive Clipping and DP-hyper are estimated values, as the actual percentages depend on specific datasets.

| Method | Searching $\eta$ | Searching $R_g$ | % Budget on Hyperparam |
|---|---|---|---|
| Vanilla Abadi et al. (2016) | ✓ | ✓ | 0% |
| Automatic Clipping Bu et al. (2023b) | ✓ | × | 0% |
| Adaptive Clipping Andrew et al. (2021) | ✓ | × | ≈20% |
| DP-hyper Papernot & Steinke (2021) | ✓ | × | ≈20% |
| Ours (this work) | × | × | <1% |

**Taylor approximation of next-iteration loss** This analysis of next-iteration loss via the Taylor approximation is also presented in Section 2 of (Bu et al., 2024b), which focuses on the explanation of the DP training dynamics, instead of on the learning rate.

