# OpenReview forum: "Towards hyperparameter-free optimization with differential privacy"
_ICLR.cc/2025/Conference — ICLR 2025 Spotlight_

### Official Review · Reviewer_2fNR · 2024-10-28

**Soundness:** 4
**Presentation:** 4
**Contribution:** 3
**Rating:** 8
**Confidence:** 3

**Summary:**

This paper proposes a method for differentially private training that eliminates the need for hyperparameter tuning, addressing a core challenge in DP deep learning. The authors provide clear discussions on the method’s efficiency, privacy guarantees, and utility. Both theoretical and empirical analyses are well-founded and straightforward.

**Strengths:**

This is a well-written paper that effectively present its methods. The motivation is clear, the connections to previous work are discussed, and the experimental results are comprehensive and convincing. The method is simple yet effective in terms of efficiency and utility. The theoretical results are presented clearly, avoiding unnecessary complications, and the experiments are solid.

**Weaknesses:**

See below.

**Questions:**

1. In Theorem 1, when $R_l \approx L$, the clipping bias is close to zero, where $L = \frac{1}{B} \sum_i L_i$ is the public per-sample gradient (right?), which seems to be a CLT-based result (please correct me if I’m wrong). My questions are:
- (1) Could the authors provide guidance on minimum batch sizes needed (to make CLT works) in practice, based on their empirical observations? Although one always wants large batch sizes but due to limited computational resources, one often can't afford super large batch sizes.
- (2) I understand why you set $\tilde{L}\_{t-1}^{(0)}$ as $R\_l$ for the next iteration, given that the loss values are similar. However, while $L\_{t-1}$ might be close to $L\_{t}$,  I worry that $\tilde{L}\_{t-1}$ could differ significantly from $\tilde{L}_{t}$ because of the clipping and noising, which might not give bias $\approx 0$. Some discussion or empirical results on this would be valuable.
- (3) What was the rationale for choosing $\tilde{L}\_{t-1}^{(0)}$ specifically? Did the authors experiment with other options like $\tilde{L}\_{t-1}^{(+1)}$ or $\tilde{L}\_{t-1}^{(-1)}$, and if so, what were the results?

2. In the main algorithm, I assumed $\eta_i \in \\{-1, 0, +1\\}$ represents the series of potential lrs. Is there a specific reason for this choice? I understand the need for at least two $\eta_i$'s, but $\{-1, +1\}$ seems more intuitive to me...? Could the authors explain the rationale behind including 0 in the set of potential learning rates? Are there specific benefits for this choice? Also, I’m unclear about how to fit eqn. (6). In Section 4.4, the authors mention that solving this is quite efficient, with the cost "mainly arising from additional forward passes." Could the authors provide more details on the practical implementation of solving equation (6), and specifically, what optimization method was used, and how much computations were typically required to find a solution?

3. Could the authors provide insights into why D-adaption and Prodigy struggle in low-$\epsilon$ regimes for full finetuning, as seen in the first table of Table 2 and Table 3? Are there specific aspects of these methods that make them less suitable for differentially private optimization? Also, for clarity, could the authors specify the value of $K$ used for HyFreeDP results in Tables 2 and 3? I assumed $K=10$ throughout these experiments, but If it varies, a note explaining the choice for each experiment would be helpful.

4. I noticed in Table 2 that NonDP-GS w/ LS outperforms HyFreeDP, especially on CIFAR-10, and in Table 3, NonDP-GS and HyFreeDP show similar performance. Do authors have any intuitions behind? I’m particularly curious why NonDP-GS w/ LS performs so well on CIFAR-10 dataset - is it because the task is too simple? If I understand correctly, NonDP-GS does not account for privacy loss from hyperparameter tuning, so the $\epsilon$ values for NonDP-GS might be underestimated. It would be great to include the results for NonDP-GS, considering the privacy cost of tuning. I imagine that HyFreeDP would then strictly outperform it...?

5. It seems to me that this method works for per-batch clipping (since it also ensures dp [1]) as well, except that eqn (7) needs to be modified. It would be particularly useful for differentially privately training models with non-deomposbale loss [1, 2].

[1] Huang, Alyssa, Peihan Liu, Ryumei Nakada, Linjun Zhang, and Wanrong Zhang. "Safeguarding data in multimodal ai: A differentially private approach to clip training." arXiv preprint arXiv:2306.08173 (2023).
[2] Kong, William, Andrés Muñoz Medina, and Mónica Ribero. "DP-SGD for non-decomposable objective functions." arXiv preprint arXiv:2310.03104 (2023).

---

> ### Author Response · Authors · 2024-11-20
>
> We thank the reviewer for the comments and all your questions are well-received! Here is our point-to-point response.
>
> **Q1-[loss clipping threshold]:** In Theorem 1, $L$ is the mean of non-privatized per-sample losses (not gradients).
>
> - (1) We empirically observe $B>100$ suffices, though this is law of large numbers rather than CLT. On the computation side, libraries like fastDP allows the same computation efficiency for DP compared to standard non-DP training. Especially, the large batch can be fitted in GPU with the gradient accumulation technique, by breaking into smaller micro batches.
> - (2) In current Figure 2 (by comparing dots to stars), we empirically demonstrate the influence of clipping and noising on each loss values (and corresponding curve fitting), which illustrates that clipping and noise have little impact.
> - (3) As we stated in Sec 4.1 (Line 267), we have tried $\sum \overset{\sim}{L}_{t-1}^{(k)}$ to further avoid clipping bias. As we can observe in Figure 2 that there is almost no gap between "w/o clipping w/o noise" and "w/ clipping w/o noise".
>
> **Q2-[choise of $\eta$ range]:**
>
> - We need at least 3 points to solve the quadratic function uniquely (with the form $ax^2 + bx + c$) in Eq (6).
> - Including $L^0$ (via $\eta_i=0$) is crucial because sampling points around $\eta_i=0$ allows the quadratic function to capture the local behavior of the loss at $w_t$ effectively.
> - Implementation for Eq(6): Given the x-list $\{-\eta, 0, \eta\}$ and the y-list $\{L^{-1}, L^0, L^{+1}\}$ for the quadratic function $y=ax^2+bx+c$, we input them into the the out-of-box function via `from scipy.optimize import curve_fit` and obtain the corresponding value for a, b, and c.
>   - Internally, it uses nonlinear least squares optimization, typically via the Levenberg-Marquardt algorithm (by default in SciPy).
>   - The operations are lightweight, which take microseconds on a decent GPU.
>
> **Q3-[Presentation]:** We stated the intuition for why methods as D-adaptation do not work in current Section 2.3. In short, these methods require some quantities to be accurately estimated that are very sensitive to the noise, especially in high dimension when model is large. We updated the $K=5$ configuration in **revision Sec 4.1**.
>
> **Q4-[Intuition on results]:** To clarify, both NonDP-GS and NonDP-GS w/ LS do not take privacy cost in hyper-parameter tuning, and should serve almost as an upper bound of performance, given that these methods trade-off the privacy guarantee. That is, $\epsilon$ in these methods are indeed underestimated. The "results for NonDP-GS, considering the privacy cost of tuning" is DP-hyper (as we introduced in Line 370).
>
> And HyFreeDP indeed strictly outperforms DP-hyper (as shown in Figure 2).
>
> **Q5-[non-deomposable]:** Thanks for the insights! We are glad that reivewer finds it useful for non-depmposable loss. We note that loss clipping in Eq(7) is general and not influenced by gradient privatization.

---

### Official Review · Reviewer_e2ge · 2024-11-01

**Soundness:** 3
**Presentation:** 3
**Contribution:** 3
**Rating:** 8
**Confidence:** 3

**Summary:**

This paper discusses how to improve the performance of DP optimization by improving the private selection of hyperparameters. This is done by always scaling the gradient to norm 1 and adjusting the learning rate for each training step privately. The selection of learning rate is done by adapting a rule used in non-DP literature which solves an approximate objective for the best learning rate, but making this private by privatizing the loss evaluation used in this learning rate objective. Extensive experiments shows this improves performance over alternative DP hyperparameter selections, and is often comparable to a non-DP grid search of hyperparameters. Further experiments show the computational overhead is minimal. Certain experimental details are missing in the text, though I believe this can be easily addressed.

**Strengths:**

1) Hyperparameter selection in DP does not inherit the same rules of thumb as non-DP, and hence understanding hyperparameter selection for DP training is a core problem
2) The results are strong and seemingly near optimal (given the non DP grid search results)
3) The observation that a specific generic ML hyperparameter selection approach translates well to the DP setting seems important and novel to the DP community

**Weaknesses:**

1) In my opinion the presentation can be improved: in the questions I try and clarify several possible typos. I emphasize this as I originally had more issues with the claims made in the paper but was able to answer these by cross-examining statements elsewhere, which could have been more easily resolved with some changes to writing.

2) Certain details about the experimental setup are missing, such as exact DP $\delta$ values used, and the range of hyperparameter grid search. I ask questions about these in the questions section (labelled Major), and believe they can be easily addressed, and am willing to increase my score given they are addressed in the rebuttal.

**Questions:**

1) The line numbers are missing in the pdf, the authors may want to fix this. In the following I will try my best to describe the location of the typo


2) In equation 6, I believe you mean to also use the privatized loss for L(w), currently it is un-privatized and hence the objective is not private. I can see from the algorithm you input to equation 6 privatized losses, so I presume this is a typo in equation 6

3) In the opening paragraph of section 4.1, I do not believe saying Auto Clipping is equivalent to Rg = 0 is correct. This would mean you always clip to 0 gradient norm. I believe you can replace this by saying you always clip to gradient norm 1, which is consistent with equation 1.

4) In algorithm 1 could you write the inputs to algorithm, which I take as initial values for: $\eta$, $R_l$. This would help with reading/understanding the algorithm

5) In line 8 of algorithm 1, can you replace $(-\eta,0,\eta)$ with set notation {$\eta, 0,\eta$} to be clearer how equation 6 is instantiated; at first I thought it was run over an interval.

6) In line 9 of algorithm 1 I find it vague as stated; more precisely can you say you minimize the previous fitted quadratic?

7) (Major) In section 5.1 experiment setup paragraph, can you explicitly state the deltas used in the experiments; I could not find this in the appendix either.

8) (Major) In table 2, can you add error bars for the experiments? E.g., report 1 standard deviation

9) (Major) Can you report the grid search range in the appendix?

10) Can you explain what BitFit is briefly in the experimental setup paragraph; I believe this will clarify better the differing results for full finetuning

11) I found the figures hard to read when I printed the paper; consider increasing the size and possibly moving one figure to the appendix.

---

> ### Author Response · Authors · 2024-11-20
>
> We thank the reviewer for the comments and all your questions are well-received! Here is our point-to-point response.
>
> **W1 & minors-[presentation]:** We greatly appreciate your efforts and we fixed every presentation issue in the Questions (Q1-Q6, Q10-Q11) in our revision. Specifically,
> - Q3: We agree that we need to both setting $R_g\to 0^+$ and enlarging $\eta \to \eta/R_g$, so as to normalize the gradients. We kindly suggest that this is different than setting $R_g\to 1$, which does not enlarge per-sample gradients whose norms are smaller than 1. We modify **Sec 4.1** to include this point.
> - Q6: We change Line 10/11 in **Algorithm 1** to help understanding the algorithm.
> - Q10: We add a brief introduction of BitFit and LoRA in the **first paragraph of Sec 5**.
>
> **W2 & majors-[experimental details]:**
>
> - Q7: We follow the standard setting to ensure $\delta<1/n$. We specifically use $\delta=n^{-1.1}$ as declared in the **first paragraph of Sec 5**.
> - Q9: We ensure the range is wide enough to cover the optimal choice inside. For clarity, we added detailed setups with references in the **second paragraph of Sec 5**.
> - Q8: Due to time limit, we first show the error bars for one set of main results in table 2 as below. We will update all cells in Table 2 in future revision. The advantage compared to previous works are still significant.
>
> | Method       | CIFAR10      | CIFAR100     | SVHN         | GTSRB        | Food101      |
> | ------------ | ------------ | ------------ | ------------ | ------------ | ------------ |
> | NonDP-GS     | 96.49 ± 0.02 | 79.7 ± 0.13  | 89.86 ± 0.21 | 67 ± 0.44    | 70.38 ± 0.23 |
> | D-adaptation | 23.74 ± 0.19 | 0.8 ± 0.01   | 15.27 ± 0.08 | 1.86 ± 0     | 1.17 ± 0.01  |
> | Prodigy      | 27.54 ± 0.61 | 0.8 ± 0.01   | 15.27 ± 0.08 | 1.86 ± 0     | 1.19 ± 0     |
> | DP-hyper     | 92.98 ± 0.14 | 74.63 ± 0.09 | 34.58 ± 0.58 | 28.23 ± 0.91 | 19.06 ± 0.59 |
> | HyFreeDP     | 96.36 ± 0.03 | 81.23 ± 0.09 | 93.24 ± 0.05 | 74.68 ± 0.54 | 74.08 ± 0.02 |
>
> Please kindly let us know if our response addresses your concerns or if there are more questions.

---

> > ### Comment · Reviewer_e2ge · 2024-11-21
> > **Response to Authors**
> >
> > I thank the authors for their response! My main concerns have been addressed and I have raised my score accordingly.

---

### Official Review · Reviewer_YG4L · 2024-11-03

**Soundness:** 3
**Presentation:** 2
**Contribution:** 3
**Rating:** 6
**Confidence:** 3

**Summary:**

The paper proposes to tune learning rate privately based on quadratic approximation during DP-SGD training. It is shown that the proposed algorithm can achieve comparable performance with non-DP learning rate search.

**Strengths:**

1. The proposed algorithm works pretty well in the experiment with not much additional cost in computation cost and privacy.
2. The idea is simple yet effective, factorizing learning rate tuning during training using quadratic approximation.

**Weaknesses:**

The proposed algorithm seems to still require a initial learning rate and the algorithm's sensitivity to the initialization seems missing.

**Questions:**

For DP hyper-parameter optimization, have the authors considered using gradients from backpropagation w.r.t learning rate to tune the learning rate privately?

---

> ### Author Response · Authors · 2024-11-20
>
> We thank the reviewer for the comments and liking this work! Here are our point-to-point response.
>
> **W1-[initial learning rate]:** We have empirically observed that an initial learning rate $1e-4$ is robust to work well for all tasks (making it a data-independent choice). Besides, we note that previous work in the non-DP regime shows that the performance is robust to initial learning rates (see Figure 14 in [Bu et al. 2024]). In the revision, we highlight such choice in **Sec 5 and Algorithm 1**.
>
> **Q1-[other approaches]:** We have considered back-propagation on $\eta$, i.e. updating the model with $\eta$ and simultaneously updating $\eta$ with gradient descent under another meta-learning rate. However, the introduction of this meta-learning rate keeps the number of hyperparameters the same and could be more sensitive to tuning than $\eta$. Therefore we opted out this idea.

---

### Official Review · Reviewer_UUab · 2024-11-03

**Soundness:** 2
**Presentation:** 2
**Contribution:** 3
**Rating:** 8
**Confidence:** 3

**Summary:**

This paper proposed a method to estimate hyperparameters (i.e., learning rate) in differentially private optimization with gradient normalization (instead of gradient clipping). As learning rate is the main tuning parameter, the proposed optimizer is hyperparameter free. The proposed additionally differentially privatizes the loss (a scalar) for estimating the learning rate.

**Strengths:**

- Setting hyperparameters in DP optimization is an important topic for both modeling and privacy.
- Experiments demonstrate the advantage of the proposed method compared to naively applying parameter free optimization methods like D-adaptation in DP optimization, and DP-hyper style algorithm by differentially privatizing hyperparameter search.

**Weaknesses:**

The technical part of the paper is generally hard to read. I am not confident the proposed method is technically correct.

**Questions:**

I have to read GeN Bu et al. (2023) again to understand the proposed method in this paper. And I cannot find Eq (5) of this draft in GeN Bu et al. (2023). What is \omega in Eq (5) and (6)? Could you write the closed form solution for estimating \eta? If not, why?

I request the authors to clarify privacy accounting of their proposed method. Starting from the DP definition, e.g., what are their mechanism input and output? How are their “Loss Privatization” and “Gradient Privatization” composed? It looks to me Line 9 in Alg 1 is data dependent, but it is unclear whether it is considered in the DP algorithm or accounting. It is OK to use GDP and/or autoDP library, but I request the authors to detail how GDP/autoDP is used for this specific algorithm.

Minor: the authors might consider the usage difference of \citep and \citet in writing.

---

> ### Author Response · Authors · 2024-11-20
>
> We thank the reviewer for the comments! We would make every effort to improve the presentation and to asssure that our method is technically correct.
>
> **Q1-[cannot find Eq(5)]:** The quadratic function in our Eq(5) is an equivalent form of Eq (2.3) in GeN [Bu et al. 2024]. We fixed and clarified the citation in the **revision Sec 2.3**, thus our paper is self-contained without the necessity to read their work.
>
> **Q1-[$w$ in Eq(5-6)]:** As we stated in Sec 2.1, $w$ is the model parameters.
>
> **Q2-[closed solution for $\eta$]:** The closed from of $\eta$ for non-DP case is written in our Eq (4). The closed form for DP case is given in **Line 173 as well as Algorithm 1 (line 11)**.
>
> **Q3-[privacy accounting]:**
> The major part of privacy accounting is elaborated in Appendix C. The “Loss Privatization” and “Gradient Privatization” are composed by existing theories, e.g. via GDP in Equation 9. To improve clarity, we append detailed pseudo code in **revision Appendix Alg 2** with detailed functions to implement.
> - For privacy accounting, as we have shown in Figure 1 with the green box, the input includes total DP privacy budget $(\epsilon, \delta)$, constant $K$ (e.g., 3), and other data-independent hyper-parameters $B, N, T$; The output includes noise magnitude for gradient and loss $\sigma_g$ and $\sigma_l$.
> - For the whole algorithm, the input is the training dataset $D$, the output is the trained model parameters $w$ and all hyper-parameters, following previous works [Papernot et al. 2021, Wang et al. 2023]. We clarified it in the revision **Sec 2.2**.
>
>   > Following the framework of previous works [Liu et al. 2019, Papernot et al.], suppose there are $m$ privacy-preserving training algorithms $\mathcal{M}_1, \cdots, \mathcal{M}_m$ which corresponds to $m$ possible hyper-parameters. Denoting the whole process of training and hyper-parameter tuning as $\mathcal{M}$, it takes input as a training dataset $D$ and outputs the best outcome over a finite set of $m$ possible hyper-parameters by running $\mathcal{M}_1, \cdots, \mathcal{M}_m$. The end-to-end DP ensures that $\mathcal{M}$ satisfies $(\epsilon, \delta)$-DP with respect to any neighboring datasets and any outcome. And the outcome includes the best model parameters and the corresponding hyper-parameters. In our hyper-parameter-free solution, there is a single training with $m=1$, and the output hyper-parameters are data-independent.
>
> - Line 9 in Alg 1 has been accounted in total privacy cost, as $\tilde{L}$ is privatized.
>
> Please kindly let us know if our response clears your concern about the correctness of our method.

---

> > ### Comment · Reviewer_UUab · 2024-11-22
> >
> > Thanks for the response and improving the clarity. My major concern on privatizing hyperparameters has been mostly addressed. I have some additional comments that hope to get feedback before the discussion phase, and happy to adjust the score again.
> >
> > 1) It is impressive that a reliable learning rate can be estimated from three points in line 9 of Alg. 1. Especially when we have DP. Maybe I missed it, could the authors comment on exactly how much noises are added, e.g., \sigma_l? It would be also great to list the gradient noise multiplier.
> >
> > 2) How are the values computed in line 10 of Alg. 1. I think there is a closed form solution, and interested in seeing it (optional for this rebuttal).
> >
> > 3) In addition to the robustness to K, it would also be nice to study the robustness to initial learning rate \eta and initial clipping R_l. This is not required for now if they are not already in the paper as we are approaching the end of the discussion.

---

> > > ### Author Response · Authors · 2024-11-25
> > > **Follow-Up Comment by Authors**
> > >
> > > Thank you again for engaging in the discussion. I hope our previous response clarified your recent questions.
> > >
> > > Please feel free to let us know if there are any remaining points you would like us to expand upon.
> > > We will do our best to address your questions promptly.
> > >
> > > We sincerely appreciate your thoughtful feedback and the time you’ve dedicated to improving our work.

---

> > > > ### Comment · Reviewer_UUab · 2024-11-26
> > > >
> > > > Thanks for the response. I may have missed it, could you explicitly list the values of hyperparameters that may affect privacy-utility trade-off? i.e.,  the value of \sigma_l, \sigma_g, batch size etc. Just trying to get a better understanding of practical implication. Not necessarily for all the experiments you run, just a typical/recommended setting would be good.

---

> > > > > ### Author Response · Authors · 2024-11-26
> > > > >
> > > > > Thanks for your question! For example, given dataset size $N=50000$, privacy budget $\epsilon=1, \delta=2e-5$, here is a set of noise related parameters. In general, the update interval $K$ controls the trade-off between more frequent adjustment ( smaller $K$) and smaller loss noise magnitude (larger $K$).
> > > > >
> > > > > And we show in Figure 4, the converged performance is robust to the choice of $K$. We by default use $K=5$ and demonstrated the cross-task robustness.
> > > > >
> > > > > | Related  parameters | K=1   | K=5  | K=10 |
> > > > > | ------------------- | ----- | ---- | ---- |
> > > > > | $\sigma_l$          | 12.12 | 5.48 | 3.92 |
> > > > > | $\sigma_g$          | 1.54  | 1.54 | 1.54 |
> > > > > | Initial $R_l$       | 1     | 1    | 1    |
> > > > > | $B$                 | 1000  | 1000 | 1000 |
> > > > >
> > > > > Furthermore, we demonstrated more noise magnitude examples in Figure 3.

---

> > > > > > ### Comment · Reviewer_UUab · 2024-11-26
> > > > > >
> > > > > > This looks reasonable, and I raised the score again.
> > > > > >
> > > > > > The epsilon=1 results still look surprisingly good. How many fine-tuning iterations? And how are the models pre-trained for vision tasks?

---

> > > > > > > ### Author Response · Authors · 2024-11-27
> > > > > > >
> > > > > > > Thank you for raising your score again!
> > > > > > >
> > > > > > > The number of fine-tuning iterations depends on the dataset size. The details are as follows:
> > > > > > >
> > > > > > > | Dataset     | Num Samples | Num Iterations | Batch Size | Epoch |
> > > > > > > | ----------- | ----------- | -------------- | ---------- | ----- |
> > > > > > > | CIFAR10/100 | 50,000      | 250            | 1,000      | 5     |
> > > > > > > | SVHN        | 73,257      | 365            | 1,000      | 5     |
> > > > > > > | GTSRB       | 39,209      | 195            | 1,000      | 5     |
> > > > > > > | Food101     | 75,750      | 375            | 1,000      | 5     |
> > > > > > > | E2E         | 42,043      | 420            | 1,000      | 10    |
> > > > > > >
> > > > > > > For vision tasks, we use the ViT models provided by the `timm` library (PyTorch Image Models) with `pretrained=True`. These models are pre-trained on ImageNet and do not require any additional pre-training in our setup.

---

> ### Author Response · Authors · 2024-11-22
>
> We appreciate the reviewer's detailed comments! Here is our response to your new concerns:
>
> Q1: As shown in **Figure 2**, the effective noise scale on the loss value is about 0.1 to 0.5, which depends on $\sigma_l, R_l$ and batch size $B$. Specifically, we demonstrated $\sigma_l$ (for loss) and $\sigma_g$ (for gradient) in **Figure 3**, with respect to different update interval $K$. Generally speaking, we recommend a relatively large K (e.g., 5) and 3 to 5 points for good fitting.
>
> Q2: The values in Line 10 Algorithm 1 can be solved by **Eq(6).** More specifically,
> - While mathematically equivalent, we opted to use `scipy.optimize.curve_fit` for better numerical stability. Given the x-list $\{-\eta, 0, \eta\}$ and the y-list $\{L^{-1}, L^0, L^{+1}\}$ for the quadratic function $y=ax^2+bx+c$, we can get (a, b, c) using `curve_fit`. Internally, it uses nonlinear least squares optimization, typically via the Levenberg-Marquardt algorithm (by default in SciPy).
> - The closed-form solution is very complicated in general for this minimization problem. Yet, we give it for three points (with details to be added in the camera-ready):
>
> $\frac{\eta}{2}\frac{\tilde{L}(w+\eta g_\text{DP})-\tilde{L}(w-\eta g_\text{DP})}{\tilde{L}(w+\eta g_\text{DP})-2\tilde{L}(w)+\tilde{L}(w-\eta g_\text{DP})}$
> where $\tilde{L}$ is the privatized loss and $g_\text{DP}$ is the privatized gradient.
>
> Q3: We appreciate your constructive suggestions. We will add experimental analysis on the robustness to $\eta_0$ and initial $R_{l}$ in our future version (may finish after discussion period). In this work, we have demonstrated the robustness by fixing $\eta_0=1e-4$ and initial $R_l=1$, then varying different models and datasets. While this is different to an experiment that fixes a model/dataset and then varies $\eta_0$ and initial $R_l$, we think the robustness is (maybe indirectly) observed and we have provided a default configuration as a guideline for practitioners.
>
> Please kindly let us know if you have more comments. We are glad to solve them all. Thanks!

---

### Public Comment · ~Ashwinee_Panda1 · 2024-12-04
**Minor Comment on Related Work**

I enjoyed reading the paper. I have a couple of minor comments on related work for DP HPO. The authors state "The more explored approach is to assign a small amount of privacy budget to privatize the hyperparameter tuning. Examples include:..." on Lines 059-060, and include a citation to one of our papers, which I appreciate. I would just like to point to a couple of previously published papers that I think are relevant.

(NeurIPS 2023) https://proceedings.neurips.cc/paper_files/paper/2023/hash/59b9582cd35f555ea8415030073e7b22-Abstract-Conference.html
(ICML 2024) https://icml.cc/virtual/2024/poster/34966 {**Disclaimer: This is our own paper**}

I understand that this paper is clearly distinct, in that the authors present a method for determining an automatic learning rate schedule. However, our ICML 2024 paper does include results for eps=1 privacy budget full finetuning of Vit-small / Vit-base on CIFAR10 and CIFAR100, so they are in some sense comparable. Although, this work clearly has lower runtime overhead.

I would also comment on the statement in Lines 431; "LoRA typically requires a LR that is 10x larger than FFT" -this is a tricky statement, because the LR in LoRA is really two hyperparameters: the alpha term and the learning rate itself. As prior work has shown (TMLR 2024 https://openreview.net/forum?id=aloEru2qCG&noteId=Jb3PQNQDI2) if alpha is scaled appropriately as alpha=2*rank, then the learning rate doesn't actually need to be much larger. I think this is probably a point that could work in favor of the paper, because as one reviewer noted, there is a hyperparameter, which is the base learning rate.

---

> ### Public Comment · ~Zhiqi_Bu1 · 2025-03-01
>
> Hi Ashwinee,
> Thank you for your comment and sharing your appreciation of our work! We agree these papers are relevant and include all of them in our camera ready. It is generally exciting to read and work on DP optimization/hyperparameter tuning. Please feel free to reach out if you would like to extend the discussion.

---

### Meta-Review · Area_Chair_nGFa · 2024-12-22

**Metareview:**

This paper discusses how to improve the performance of DP optimization by improving the private selection of hyperparameters. This is done by always scaling the gradient to norm 1 and adjusting the learning rate for each training step privately. The selection of learning rate is done by adapting a rule used in non-DP literature which solves an approximate objective for the best learning rate, but making this private by privatizing the loss evaluation used in this learning rate objective. Extensive experiments shows this improves performance over alternative DP hyperparameter selections, and is often comparable to a non-DP grid search of hyperparameters. Further experiments show the computational overhead is minimal. Certain experimental details are missing in the text, though I believe this can be easily addressed.

All the reviewers agree that this paper has enough novelty and contribution to be published at ICLR.

**Additional Comments On Reviewer Discussion:**

The reviewers and authors actively engaged in fruitful discussions. The outcomes are positive.

---

### Decision · Program_Chairs · 2025-01-22

Accept (Spotlight)